# Autologous Peripheral Blood Mononuclear Cells in Patients with Small Artery Disease and Diabetic Foot Ulcers: Efficacy, Safety, and Economic Evaluation

**DOI:** 10.3390/jcm12124148

**Published:** 2023-06-20

**Authors:** Benedetta Ragghianti, Bianca Maria Berardi, Edoardo Mannucci, Matteo Monami

**Affiliations:** Department of Diabetology, Careggi University Hospital, 50121 Florence, Italy; b.ragghianti@gmail.com (B.R.); biancamaria.berardi@stud.unifi.it (B.M.B.); edoardo.mannucci@unifi.it (E.M.)

**Keywords:** diabetes mellitus, foot ulcer, cell therapy, small artery disease, chronic limb-threatening ischemia, economic evaluation

## Abstract

Background: diabetic foot ulcers (DFU) represent the main cause of major amputations and hospitalisations in diabetic patients. The aim of this study was to assess the safety and cost-efficacy of intramuscular injection of peripheral blood mononuclear cells (PBMNCs) in diabetic patients with no-option chronic limb-threatening ischemia (CLTI) and small artery disease (SAD). Methods: a retrospective study was carried out on a series of type 2 diabetic patients with DFU grade Texas 3 and no-option CLTI and SAD. All patients had undergone at least a previous revascularization and were allocated to a surgery waiting list for major amputation. The principal endpoint evaluated at 90 days was a composite of TcPO_2_ values at the first toe ≥30 mmHg and/or TcPO_2_ increase of at least 50% from baseline and/or ulcer healing. Secondary endpoints were individual components of the primary endpoint, any serious and non-serious adverse events, and direct costs at one year. Results: the composite endpoint was achieved in nine patients (60.0%); one patient (6.7%) healed within ninety days and 26.7% and 46.7% showed TcPO_2_ ≥ 30 mmHg and a TcPO_2_ increase of at least 50% at ninety days, respectively. At one year, three (20.0%) patients underwent a major amputation (all diagnosed SAD grade III). One patient died after seven months, and seven patients (46.7%) healed. The overall median and mean cost per patient were EUR 8238 ± 7798 and EUR 4426 (3798; 8262), respectively. Conclusions: the use of PBMNCs implants in no-option CLTI diabetic patients with SAD seems to be of help in reducing the risk of major amputation.

## 1. Introduction

DFUs (diabetic foot ulcers) represent the main cause of major amputations and hospitalisations in diabetic patients [1]. The major amputation rate in diabetic patients is 15 times superior to that of non-diabetic patients [2] and about 85% of limb amputations are preceded by a foot ulcer [3]. The most important risk factors for the development of DFUs are diabetic neuropathy and peripheral artery disease, which are frequently concomitant [4].

Chronic limb-threatening ischemia (CLTI), affecting about 25% of diabetic patients [5], represents the most advanced form of peripheral artery disease (PAD), responsible for a considerably higher rate of major amputation [6] and mortality [7].

The gold standard for the treatment of CLTI is percutaneous or surgical revascularization. However, up to 25% of diabetic patients with CLTI are not eligible for revascularization due to technical difficulties in overcoming vessel obstruction and/or a high number of comorbid conditions [8,9]. CLTI is defined as ‘no-option’ ischemia in cases of the absence of a suitable target arterial path with no visible distributing arterial circulation in the foot (“desert foot”) [10].

Diabetic patients with no-option CLI (NO-CLTI) are at higher risk of major amputation (30% vs. 4.5%, *p* = 0.0001) and mortality (50% vs. 8.9%, *p* < 0.0001) in comparison to patients undergoing revascularization [9]. This risk is even higher in the presence of small artery disease (SAD), which is an often-neglected condition affecting patients with diabetes and/or renal insufficiency and dialysis [11]. SAD is a complex vascular disorder defined as a disease of the small vessels of the plantar arch [11]. Despite its relevant prevalence and clinical significance, current therapeutic options for SAD are limited and often ineffective, leading to high morbidity, amputation, mortality, and direct and indirect healthcare costs.

In recent years, cell therapy has emerged as a promising approach to addressing NO-CLTI by promoting angiogenesis, vasculogenesis, and tissue repair. Autologous cell therapy, in fact, has shown favourable effects on several outcomes, such as pain, transcutaneous oxygen tension, ulcer healing, major amputation, and mortality [12]. Stem cells increase peripheral circulation by stimulating neo-angiogenesis achieved through paracrine activities of growth factors, cytokines, and messenger molecules, as well as through exosomes [13,14].

Cell therapy (i.e., mesenchymal stem cells and blood marrow mononuclear cells) can be delivered through different routes and methods depending on the cell type, stage of the disease, and treatment goal. The most common routes of administration include intramuscular injection, intravenous infusion, direct injection into the target tissue or muscle, and delivery through a biomaterial or scaffold [15,16]. In recent years, some authors have proposed the intramuscular injection of peripheral blood mononuclear cells (PBMNCs), which has shown similar efficacy in comparison with “traditional” autologous stem cells. Notably, this new approach presents several advantages, such as less invasive extraction techniques not requiring hospitalisation, less painful and time-consuming procedures, etc. [15,16].

No data on the efficacy and safety of cell therapy for diabetic patients with SAD have been published so far, and therefore the present retrospective study is aimed at evaluating the cost-effectiveness and safety of the PBMNCs implant in diabetic patients with no-option CLTI and SAD allocated to a surgery waiting list for a major amputation.

## 2. Patients and Methods

The present analysis was performed on a consecutive series of NO-CLTI patients with DFUs and SAD who underwent the implantation of PBMNCs from peripheral blood at the Diabetic Foot Unit of Careggi Hospital, Florence, Italy, between 1 January 2020 and 30 June 2021. All patients were candidates for elective major amputations, and allocated to a surgery waiting list.

The study protocol was approved by the local ethical committee (Protocol number SPE_22580) and informed consent was obtained from all patients before the inclusion in the analysis.

Patients were included if fulfilling the following criteria:(1)Diagnosis of diabetes mellitus;(2)Age > 18 years;(3)DFUs grade Texas 3;(4)No-option CLTI and SAD (see below for definitions);(5)Allocation to a surgery waiting list for major amputation;(6)At least one previous revascularization procedure (endoluminal or open surgery);(7)Absence of severe infection according to the PEDIS classification system (PEDIS < 2; [17]);(8)Absence of severe anaemia (Hb > 8 g/dL);(9)Absence of coagulation disorder/thrombocytopenia (PLT > 50,000/L);(10)Absence of active cancer/leukaemia or lymphoma or haematological disease;(11)Being able to sign informed consent.

CLTI was diagnosed in cases of ischemic pain at rest or ischemic ulcer/gangrene at foot level associated with systolic blood pressure at ankle level <70 mmHg or systolic blood pressure at first toe <50 mmHg or TcPO_2_ values at foot level <30 mmHg [18].

SAD was defined according to a global evaluation of the arch and the small foot arteries as:Grade 1:patent: absence of disease or mild disease with a well-represented network of forefoot and calcaneal arteries;Grade 2:stenosis (or mild disease): diffuse disease with narrowing and poverty of metatarsal, digital and calcaneal arteries;Grade 3:occlusion (or severe disease): extreme poverty of arch, metatarsal, digital and calcaneal arteries.

The above as defined by the evaluation of two operators and described in the paper of Ferraresi et al. [13]. A vascular surgeon and an interventional cardiologist confirmed the presence of SAD disease by reviewing all angiographic procedures. SAD was defined according to a global evaluation of the arch and the small foot arteries (including calcaneal branches, tarsal, metatarsal and digital arteries) as grade 1: absence of disease or mild disease with a well-represented network of forefoot and calcaneal arteries; grade 2: diffuse disease with narrowing and poverty of metatarsal, digital and calcaneal arteries; and grade 3: extreme poverty of arch, metatarsal, digital and calcaneal arteries [11].

All patients received a multidisciplinary evaluation by vascular surgeons and interventional cardiologists to explore the possibility of a new lower-limb revascularization. Patients were therefore included in the present analyses only if considered (1) not eligible for a new revascularization according to ESVS-ESC 2017 criteria [18], or (2) in cases of no run-off pedal vessels or (3) failure after infra-genicular bypass grafting. The indication to perform an implantation of perilesional and perivascular monocytes was discussed collegially by the multidisciplinary team (diabetologist, vascular surgeon, and interventional cardiologist) as a limb-salvage attempt.

### 2.1. Baseline Data Collection

Demographic and clinical data were collected from clinical records, including a medical history with detailed information on the duration of diabetes, complications and concomitant medical conditions, current pharmacological treatment, cardiovascular risk factors, self-reported smoking habits and any other relevant medical condition. At the first visit, following an established standard procedure of the clinic, all patients underwent a physical examination, during which their weight, height, and blood pressure were recorded. All patients gave a blood sample after a minimum 8 h fasting (i.e., HbA1c, glycemia, creatinine, total cholesterol, HDL-cholesterol, triglycerides, transaminase, bilirubinemia, ɣ-GT, potassium and sodium).

Ulcer dimensions were evaluated with MolecuLight i:X^®^. When more than one lesion was present, only the largest ulcer was taken into account. Diagnosis of diabetic neuropathy was performed measuring the vibratory perception threshold with a biothesiometer (METEDA, San Benedetto del Tronto, Italy) and monofilament testing 10g. Ulcers were classified according to the University of Texas score [16].

Pain at the first visit and quality of life were assessed using a visual analogue scale (VAS) ranging from 0 to 10 (VAS for pain) and from 0 to 100 (VAS for quality of life), respectively.

The number of previous surgical and percutaneous homolateral revascularizations were registered.

As per local standard of care, the transcutaneous pressure of oxygen (TcpO_2_; Radiometer Medical ApS; Brønshøj, Denmark) at the base of the first toe and at the ankle (in the area of perfusion of the posterior tibial artery) and the ankle–brachial index (ABI; or toe– brachial index) were measured, and an echo colour Doppler examination of lower-limb arteries was performed.

Renal failure was defined as a reported previous diagnosis of renal failure, or as serum creatinine >1.5 mg/dL. Ischemic heart disease (IHD) and cerebrovascular disease were diagnosed when patients reported previous myocardial infarction/angina or a stroke/transient ischemic attack. Comorbidity was assessed through the calculation of Charlson’s comorbidity score (CCS).

### 2.2. Ulcer Treatment

All patients received the same standard therapy according to IWGDF guidelines [4]: surgical debridement, local dressings and foot offloading, antiplatelet drugs, antibiotic therapy in case of infection and pain relief therapy.

All patients underwent a procedure of local infiltration of autologous mononuclear cells through multiple perilesional and intramuscular injections of 10 mL PBMNCs cell suspensions (0.2–0.3 mL in boluses) performed below the knee along the relevant vascular axis (anterior tibial artery and posterior tibial artery) at intervals of 1–2 cm and to a mean depth of 1.5–2 cm, using a 21 G needle. Intramuscular injections were performed along the occluded below-the-knee vessel(s) (irrespective of the “wound-related artery”) and along the main foot vessels (such as the pedal artery and/or medial and plantar arteries), irrespective of the presence of vascular stenosis/occlusion and the wound angiosome area. All the patients received local perilesional administration.

The procedures were performed according to the instructions of the manufacturer and were repeated at least two times for each patient, at intervals of thirty days. All procedures were performed in an operating room with anesthesiologic support (midazolam iv and/or peripheral block). For these procedures, Athena Monocells Solution kits were used, following the manufacturer’s instructions [19]. Table 1 reports specific details for each included patient.

### 2.3. Follow-Up Data

Patients were evaluated at baseline and at one, three, six and twelve months after the first implantation, and the following parameters were recorded:(1)TcPO_2_ at the first toe;(2)Pain (using VAS scale from 0 to 10);(3)Vital status of patient;(4)Healing rate;(5)Major amputation rate.

After 6 months:(1)Quality of life.

### 2.4. Endpoints

The primary endpoint of the study was a composite of the following items at 90 days:-TcPO_2_ at the first toe ≥30 mmHg and/or-increase of at least 50% of TcPO_2_ in comparison with baseline values and/or-healing of the ulcer.

A 90-day time horizon was chosen to evaluate the effects of PBMNCs on TcPO_2_ after 1 month from the last procedure.

The cut-off of 30 mmHg was used, because it is well recognized as the threshold for CLTI [10].

The increase of at least 50% of TcPO_2_ in comparison with baseline values was chosen as an arbitrary cut-off for a “clinically significant” amelioration of limb perfusion.

Ulcer healing can be considered a good proxy for the success of cell therapy.

Secondary outcomes evaluated at each time point were:-Individual components of the primary endpoint;-Any serious and non-serious adverse events;-Direct costs at one year.

Complete healing was defined as full epithelialization of the wound (also obtained after minor amputation) confirmed after 7 days. Minor amputations were performed, as recommended by international guidelines [4] only with distal TcPO_2_ ≥ 30 mmHg or in cases of a 50% increase of TcPO_2_ compared with basal values; minor amputation was considered as limb rescue, and was defined as any amputation performed below the ankle. Major amputation was defined as a surgical procedure performed above the ankle.

### 2.5. Economic Assessment

The economic assessment was performed considering the perspective of the local health system, thus considering only direct healthcare costs and including costs associated with healthcare resources used throughout the follow-up and extracted from clinical records. In detail, direct costs included specialist visits, diagnostic procedures, hospital admissions (related to diabetic foot), major and minor amputations, antibiotic therapy, grafts, and off-loading orthesis. Costs for hospitalisations were estimated on the basis of established regional tariffs (https://www.salute.gov.it/portale/temi/p2_6.jspid=3662&area=programmazioneSanitariaLea&menu=vuoto (accessed on 1 April 2023)), i.e., tariffs established for the diagnosis-related group (DRG) associated with each episode for hospital admissions (either day hospital or full-length stay) and recorded in clinical records, and similarly for costs related to specialistic visits and outpatient procedures performed (e.g., RX, MRI, laboratory exams, etc.). The cost of antibiotic therapy was estimated considering ex-factory prices (https://www.salute.gov.it/portale/temi/p2_6.jsp?id=3662&area=programmazioneSanitariaLea&menu=vuoto (accessed on 1 April 2023)), while current market prices were used to value costs for orthopaedic shoes/orthesis. The health economic analysis performed attempted to estimate costs borne by the healthcare system, mainly using tariffs related to different healthcare services, over one year. As discounting typically requires the collection of data over different time points to give different values to both costs and health outcomes that are predicted to occur in the future, because they are usually valued less than the present costs, given the time frame considered in our analysis we decided not to apply any discount rate. All costs were referred to 2020, and are reported in Appendix A.

### 2.6. Statistical Analyses

Statistical analysis was performed on SPSS 25.0. Data were expressed as mean ± standard deviation (Std. dev), or as median (25th–75th percentile), depending on their distribution. Comparisons between groups were performed using the Student’s t-test for independent samples or the Mann–Whitney U test as appropriate. The chi-square and Fisher exact tests were used for between-group comparisons of categorical variables, as appropriate. The Kaplan–Meier method was used to derive the probability of healing over time.

## 3. Results

The whole cohort was composed of 15 patients (4 women, 26.7%), aged 69.8 ± 13.0 years, and affected by ischemic DFU. The principal characteristics of the patients are summarised in Table 2.

Most DFU involved the forefoot (80%), and gangrene was present in 33% of cases; the median TcPO_2_ at the first toe level at baseline was 3.8 (1.2; 22.1) mmHg and SAD grade 2 and 3 was detected in ten and five patients, respectively.

The primary 90-day composite endpoint was achieved in nine patients (60.0%). One patient (6.7%) healed within 90 days and four (26.7%) and seven (46.7%) showed TcPO_2_ > 30 mmHg and/or a TcPO_2_ increase of at least 50% from baseline, respectively. No patients underwent major amputation in the first three months of follow-up.

Median values of TcPO_2_ (at the basis of the first toe) at baseline and at 1, 3, 6, and 12 months are reported in Table 3; a significant increase in TcPO_2_ values was observed at 3, 6, and 12 months (Table 3) from baseline. For the only two patients with a hind-foot ulcer, TcPO_2_ values at the ankle level were analyzed, with a trend toward an increase in TcPO_2_ values at any time point. A statistically significant reduction in pain was observed at any time point, and quality of life measured at six months showed a nonsignificant trend toward increase.

At 1 year, three (20.0%) patients underwent a major amputation (all diagnosed SAD grade III). One patient died after seven months, and seven patients (46.7%) healed (four after minor amputations) within twelve months.

Following our internal protocol all patients except four underwent two infiltrations of PBMNCs; one patient received three infiltrations due to an incomplete response to the treatment, and the other three patients underwent major amputation before undergoing the second infiltration for clinical reasons.

No major adverse events were observed during follow-up, and only four patients reported pain immediately after the procedure (median value 3.5), which completely disappeared in a few minutes without requiring any treatment.

A formal analysis of the direct costs sustained during the 1-year follow-up is reported in Table 4. The overall median and mean cost per patient were EUR 8238 ± 7798€ and EUR 4426 (3798; 8262), respectively, which were significantly (*p* < 0.001) lower than (direct) costs which would have been sustained for major amputation (EUR 21,065).

## 4. Discussion

Chronic limb-threatening ischemia is a challenging condition for clinicians involved in the treatment of DFU. The therapeutic approach to CLTI depends on several factors such as patient-specific vascular anatomy, availability of vascular conduits for revascularization, and comorbid conditions, such as cardiac disease and renal insufficiency [8,10,11]. Peripheral artery disease in patients affected by diabetes is characterised by multisegmental distribution and distal involvement of the artery at the foot level. In these conditions, traditional endovascular techniques as well as open revascularization procedures are frequently less effective than in nondiabetic patients [20]. Moreover, revascularization procedures in diabetic patients are also challenging, due to technical reasons (e.g., absence of an autologous venous conduit for bypass or lack of a suitable pedal or plantar artery target, intima–media calcification, etc.) [20]. Some preliminary experience has shown potential additive effects of cell therapy and peripheral revascularization in diabetic patients’ recalcitrant foot ulcers [16]; however, high costs and the lack of randomized control trials prevent a wide use of a combined therapy.

Moreover, diabetes and renal insufficiency (often co-exiting) are independent risk factors for SAD, which is a further condition limiting the feasibility and efficacy of revascularization. In this complex scenario, a non negligible fraction of type 2 diabetic patients is at high risk of no-option CLTI and major amputation.

Since there is no definitive treatment for SAD, and existing therapies such as lifestyle modifications, pharmacotherapy, and revascularization procedures have limited efficacy and significant side effects, there is a growing interest in the potential use of cell-based therapies [12]. However, to our knowledge, no studies have been performed in patients (candidates for major amputation) with diabetic foot ulcers and SAD.

Despite the growing interest and the present preliminary results on the potential of cell-based therapies in SAD, there are several challenges that still need to be addressed to optimise their safety and efficacy, including the selection of the most appropriate cell type, the dose, and the delivery method, as well as the optimization of the therapeutic window, the timing, and endpoints. In addition, there are several concerns regarding the safety and immunogenicity of allogeneic cell products, the potential risk of tumorigenesis or ectopic tissue formation, and the regulatory and ethical issues related to the manufacturing, labelling, and approval of cell therapy products [21].

The present study made an attempt at verifying the affordability of this new relatively cheap automated cell processing system, developed to be used at the patient’s bedside or in the operating room [22,23]. In our study, the beneficial effects on pain, TcPO_2_, and the avoidance of major amputations in a large fraction of included patients (all allocated to a surgery waiting list for major amputation) seem to be affordable if compared with the costs sustained by other similar samples of patients with ischemic grade 3 Texas diabetic foot ulcers [24].

Our study, therefore, although limited by its retrospective nature (i.e., uncontrolled study) and the small sample size, can provide some insights into this topic and be of help for clinicians involved in the treatment of NO-CLTI patients with SAD. In fact, the obtained results (i.e., the increase in TcPO_2_ values, the reduction of pain and the avoidance of major amputation in a large fraction of patients) are encouraging and of help as a hypothesis-generating research study. In addition, there is no post-procedural angiogram evaluation to assess the potential improvement in SAD after cell-based therapy. In the present study, we have also assessed the direct costs sustained for the treatment of these patients, which are relevant but significantly lower than those needed for major amputations, and are avoided in a large fraction of patients included in the present analysis.

## 5. Conclusions

In conclusion, despite these preliminary promising results on several health outcomes (i.e., pain, TcPO_2_, and major amputation), further studies (in particular, randomised controlled trials), are needed to elucidate the mechanisms, optimise the procedures, assess the cost-effectiveness and validate the safety and efficacy of cell-based therapies for NO-CLTI complicated by SAD.

## Figures and Tables

**Table 1 jcm-12-04148-t001:** Data on specific details on PBMNCs procedures for each patient included.

Patient **	Age	Gender	# Previous PTA	Number of PBMNCs	TcPO_2_ * Baseline	TcPO_2_ at 1 Month	TcPO_2_ at 3 Months	TcPO_2_ at 6 Months	TcPO_2_ at 12 Months	Surgical Procedure
FE	76.6	M	2	1	1.2	2.4	- ^§^	- ††	- ††	TFA
MV	68.7	M	2	1	0.1	2.0	- ^§^	- ††	- ††	TTA
GN	78.4	M	1	2	25.7	30.2	36.7	57.5	39.7	TMA
UC	85.0	M	3	2	22.7	21.5	24.0	25.7	22.1	TMA
PA	50.1	M	1	1	22.1	23.1	- ^§^	- ^§^	20.8	TA
GC	39.0	W	1	2	14.0	36.0	32.9	27.0	30.0	TA
PC	57.5	W	5	2	3.8	1.0	14.5	16.1	- ^§^	-
VA	78.7	M	1	2	2.0	2.0	25.0	25.0	25.4	-
AS	74.3	W	1	2	0.2	1.0	2.0	44.5	- †	-
SD	77.6	M	4	2	0.1	0.3	1.1	26.9	- ††	TFA
GC	80.5	M	1	2	27.9	59.6	49.8	60.1	54.6	-
EDG	76.5	M	3	2	18.0	8.0	16.0	22.0	5.0	-
MA	63.0	M	4	2	15.6	26.3	17.7	55.0	26.2	-
LG	64.1	M	2	3	3.0	5.4	1.7	13.0	13.0	TA
VM	77.7	W	1	2	1.5	0.5	30.0	12.4	21.1	TA

* At the first toe; ^§^ patients missed the planned visit; † deceased after 7 months; †† Major amputation; #: means number of; ** Patients’initial (name and surname); TFA: Trans-femoral amputation; TTA: Trans-metatarsal amputation; TA: Toe amputation; PTA: Percutaneous transluminal angioplasty; TMA: Transmetatarsal amputation.

**Table 2 jcm-12-04148-t002:** Main anthropometric and demographic characteristics of the enrolled cohort and of observed ulcers.

	Case (*n* = 15)
Age (years)	69.8 ± 13.0
Gender (women, %)	4 (26.6%)
Body Mass Index (kg/m^2^)	25.4 ± 4.2
Diabetes type 2 (%)	14 (93.3)
Diabetes duration (years)	28.2 ± 10.6
Medical history and risk factors (*n*, %)	
Diabetes mellitus type 1	1 (6.6%)
Charlson’s score index	6.0 [3.0–7.0]
Peripheral artery disease	15 (100.0%)
Neuropathy	15 (100.0%)
Retinopathy	6 (40.0%)
Chronic renal insufficiency	9 (60.0%)
Dialysis	1 (6.7%)
Ischemic heart disease	10 (66.7%)
Heart failure	4 (26.7%)
Ictus	2 (13.3%)
Charcot disease	4 (19.0%)
Connective tissue diseases	2 (13.3%)
Malignancies (<5 years)	1 (6.7%)
Cognitive impairment	2 (13.3%)
Smokers	1 (6.6)
Laboratory parameters	
HbA1c (%)	57.7 ± 14.6
Creatinine (mg/dL)	1.09 [0.86; 1.59]
LDL-Cholesterol (mg/dL)	58.7 ± 34.9
Pharmacological treatment (*n*, %)	
Insulin	11 (73.3%)
Glucose-lowering agents	15 (100%)
Antiaggregants	11 (73.3%)
Anticoagulants	7 (46.7%)
Statins	15 (100%)
Main ulcers’ characteristics	
Duration (days)	365 (114; 546)
Site	
Forefoot	12 (80.0)
Midfoot	1 (6.6)
Hindfoot	2 (13.3)
Median area (cm^2^)	2.8 (0.7; 11.9)
TEXAS (*n*, %)	
3B	5 (33.3)
3D	10 (66.7)
Gangrene (%)	5 (33.3)
Osteomyelitis (%)	12 (80.0)
TcPO_2_ (at the 1st toe level, mmHg)	3.8 (1.2; 22.1)
SAD (*n*, %)	
1	0 (0.0)
2	10 (67.7)
3	5 (33.3)
Pain (VAS 0–10)	5.0 (3.0; 8.0)
Quality of life (VAS 0–100)	50 (27; 60)
Number of previous revascularizations (%)	
1	7 (46.6)
2	3 (20.0)
3	2 (13.3)
4	2 (13.3)
5+	1 (6.6)

**Table 3 jcm-12-04148-t003:** Median values of distal (at the level of first toe) TcPO_2_, perceived pain and quality of life at 0, 1, 3, 6 and 12 months. Patients with hind-foot ulcer have been excluded from TcPO_2_ analysis.

Month	0	1	3	6	12
Distal TcPo2	3.8 [1.2; 22.1]	5.4 [1.0; 26.3]	20.2 [5.1; 32.2] *	26.0 [16.4; 52.4] *	24.1 [19.0; 32.4] *
Pain	5 [3; 8]	3 [0; 6] *	0 [0; 3.5] *	0 [0; 2] *	0 [0; 0.2] *
Quality of life	50 [27; 60]	-	-	60 [30; 70]	-

* *p* < 0.05 from baseline.

**Table 4 jcm-12-04148-t004:** Average costs during the follow-up of 1 year.

	Mean Std. Dev.	Median [Interquartiles]
Minor amputations/grafts	374 ± 562	0 [0; 731]
HA for FRP	1705 ± 2508	0 [0; 4904]
Outpatient visits and laboratory exams	571 ± 261	563 [324; 780]
Major amputations	4213 ± 8722	0 [0; 0]
Antibiotics	272 ± 976	0 [0; 24]
PBMNCs	3240 ± 1009	3600 [1800; 3600]
Total costs	8238 ± 7798	4426 [3798; 8262]

HA: hospital admission; FRP: foot-related problems; Std: Standard; dev: deviations.

## Data Availability

Not applicable.

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
