# Peer review of "Autologous Peripheral Blood Mononuclear Cells in Patients with Small Artery Disease and Diabetic Foot Ulcers: Efficacy, Safety, and Economic Evaluation"

_jcm, 2023, doi:10.3390/jcm12124148_

Round 1
Reviewer 1 Report
Comments
Page 3 Line 101
“SAD was defined according to a global evaluation of the arch and 101 the small foot arteries as grade 1: absence of disease or mild disease with a well-represented network of forefoot and calcaneal arteries; grade 2: diffuse disease with narrowing and poverty of metatarsal, digital and calcaneal arteries; grade 3: extreme poverty of arch, metatarsal, digital and calcaneal arteries
How was” poverty” and “extreme poverty” defined?
Page 4 Line 152
“ The procedures were performed according to the instructions of the manufacturer and were repeated at least two times for each patient at intervals of 30 days”.
Could specific details be given for each patient?
Page 4 Line 168
“The primary endpoint of the study was a composite of the following items at 90 days: - TcPO2 at the first toe ≥30 mmHg and/or - increase of at least 50% of TcPO2 in comparison with baseline values and/or - healing of the ulcer
What is the scientific basis of this end point?
Page 6 Line 224
“Median values of TcPO2 (at the basis of the first toe) at baseline, 1, 3, 6, and 12 months are reported in Table 2; a significant increase of TcPO2 values were observed at 3, 6, and 12 months (Table 2) from baseline.”
This should be Table 3
Also, with only 15 patients, could a Table of the results for each patient at each time line be reported?
Author Response
Page 3 Line 101
“SAD was defined according to a global evaluation of the arch and the small foot arteries as grade 1: absence of disease or mild disease with a well-represented network of forefoot and calcaneal arteries; grade 2: diffuse disease with narrowing and poverty of metatarsal, digital and calcaneal arteries; grade 3: extreme poverty of arch, metatarsal, digital and calcaneal arteries. How was” poverty” and “extreme poverty” defined?
Definition of SAD has been modified for better clarity.
Page 4 Line 152
“The procedures were performed according to the instructions of the manufacturer and were repeated at least two times for each patient at intervals of 30 days”.
Could specific details be given for each patient?
A table reporting data for each patients has been now added (New Table 1).
Page 4 Line 168: “The primary endpoint of the study was a composite of the following items at 90 days: - TcPO2 at the first toe ≥30 mmHg and/or - increase of at least 50% of TcPO2 in comparison with baseline values and/or - healing of the ulcer. What is the scientific basis of this end point?
1) A 90-day time horizon was chosen to evaluate the effects of PBMNCs on TcPO2 after 1 month from the last procedure.
2) The cut-off of 30 mmHg was used, because it is well recognized as the threshold for CLTI (10).
3) The increase of at least 50% of TcPO2 in comparison with baseline values was chosen as an arbitrary cut-off for a “clinically significant" amelioration of limb perfusion.
Ulcer healing can be considered a good proxy for the success of cell therapy.
These considerations were now added in the Method section (lines 185-191).
Page 6 Line 224
“Median values of TcPO2 (at the basis of the first toe) at baseline, 1, 3, 6, and 12 months are reported in Table 2; a significant increase of TcPO2 values were observed at 3, 6, and 12 months (Table 2) from baseline.”
Done.
Also, with only 15 patients, could a Table of the results for each patient at each time line be reported?
A table reporting data for each patients has been now added (New Table 1).
Reviewer 2 Report
The article is presenting a very interesting topic and a novel method for treatment of DFU in patients with small artery disease and no option chronic limb threatening ischemia.
The topic is very relevant in the field and it addresses a specific gap in the field.
The limitation of the study is not just retrospectivity but also the lack of the control group. Perhaps they could compare the results with similar patients who were treated without topical application of autologous peripheral blood mononuclear cells in the same time period? This would significantly improve the quality of the manuscript. In such form it is difficult to evaluate the proposed method.
Authors should improve the discussion which is rather short and add some more citation from this field if they are available. Also, conclusions should be summarised.
It seems that the new method is promising but the prospective, randomised study is needed as they already stated.
Good side of the manuscript is that they have also assessed direct costs for the treatment of patients with described pathology.
References are appropriate.
Article needs major revision.
Author Response
The article is presenting a very interesting topic and a novel method for treatment of DFU in patients with small artery disease and no option chronic limb threatening ischemia.
The topic is very relevant in the field and it addresses a specific gap in the field.
The limitation of the study is not just retrospectivity but also the lack of the control group. Perhaps they could compare the results with similar patients who were treated without topical application of autologous peripheral blood mononuclear cells in the same time period? This would significantly improve the quality of the manuscript. In such form it is difficult to evaluate the proposed method.
We completely agree with the reviewer, that we thank for this precious suggestion (initially we tried to design this retrospective study as a propensity score matched study). However, a case-control study is quite hard to be performed in this case. In fact, in patients with no-option CLTI is quite unusual to routinely repeat TcPO2 after 1, 2, 3, and 6 months (as done for the “case group” to verify the efficacy of PBMNCs). Since the primary endpoint is essentially based on this parameter, data on TcPO2 would have not been available for control group patients. In addition, EC approved this study as a consecutive series of patients who received this treatment, therefore changing the design without informing EC is not allowed.
Authors should improve the discussion which is rather short and add some more citation from this field if they are available. Also, conclusions should be summarised.
Discussion was modified and some new references added. A new par. “Consclusions“ has now been added at the endo of the manuscript.
It seems that the new method is promising but the prospective, randomised study is needed as they already stated. Good side of the manuscript is that they have also assessed direct costs for the treatment of patients with described pathology.
Thank you very much for your note.
References are appropriate.
Thanks.
Reviewer 3 Report
Congratulation to authors for this manuscript. The topic is very interesting opening new perspectives for patient with NO-CLI and small artery disease.
The text is clear for readers, the methodology adequate and results promising.
I have some doubt, comments, and suggestions.
Title:
· The title may be controversial, authors speak on topical application of autologous cell therapy (ACT), actually the application is this population was intramuscular and/or perilesional and not topical. Consider to adjust the title if you retain.
Introduction:
· Authors state that “SAD is a complex vascular disorder defined as a disease of small vessels of plantar arch”. Actually, SAD is defined as the disease of small artery od the foot including calcaneal branches, tarsal, metatarsal, digital arteries and plantar arch (and not only the plantar arch). Please revise the definition.
Methodology:
· Authors report that “All patient underwent a procedure of local infiltration of autologous mononuclear cells through multiple perilesional and intramuscular injections performed below the knee along the relevant vascular axis (anterior tibial artery and posterior tibial artery)”.
May you better explain the application and technique for readers? Did you inject ACT along the occluded below-the-knee vessel(s) or only along the “wound related artery”? Did you inject ACT also in the main foot vessels such as pedal artery and/or medial and plantar arteries according to the wound angiosome area? Did you perform local perilesional administration in all patients included?
· If Authors have also tested ABI at the baseline, may you report the main values at the enrolment?
· It has been reported that TcPO2 is recorded at the baseline on the 1st toe and at the ankle (do you mean in rearfoot in the area of perfusion of posterior tibial artery?). In addition, TcPO2 was evaluated during the follow-up only on the 1st toe.
May you better explain your procedure? Why did you perform TcPO2 also at the ankle if t80% of patients had forefoot DFUs? or you mean only for patients with heel ulcers?
During the follow why did you perform TcPO2 only on the first toe also for patients with heel uclers?
Results
· In the text you reported that TcPO2 values are reported in table 2 and cost analysis in table 3. Please revise, TcPO2 values are now reported in table 3 and costs in table 2.
· In table 1, could you report the ulcers size at the baseline?
· Could you report what kind of surgical interventions the patients underwent after ACT? i.e toe amputation, transmetatarsal amputation, Lisfranc or Chopart amputation, sequestrectomy, calcanectomy, etc. It can help to understand the severity of DFUs and the burden of surgical procedures.
· Could you report (again in table 1) how many patients had a desert foot with the absence both of pedal and plantar arteries?
Discussion
· Some limitation should be added to the text such as the absence of a control group which probably is the main limitation. In addition, there is not post-procedural angiogram evaluation (not ethical of course) to evaluate the real and the grade of improvement of SAD after ACT. Nonetheless, this element does not influence the validity of the study.
Author Response
Title:
The title may be controversial, authors speak on topical application of autologous cell therapy (ACT), actually the application is this population was intramuscular and/or perilesional and not topical. Consider to adjust the title if you retain.
We agree with the reviewer’s criticism. The title has been now modified.
Introduction:
- Authors state that “SAD is a complex vascular disorder defined as a disease of small vessels of plantar arch”. Actually, SAD is defined as the disease of small artery od the foot including calcaneal branches, tarsal, metatarsal, digital arteries and plantar arch (and not only the plantar arch). Please revise the definition.
Done (see line 102)
Methodology:
- Authors report that “All patient underwent a procedure of local infiltration of autologous mononuclear cells through multiple perilesional and intramuscular injections performed below the knee along the relevant vascular axis (anterior tibial artery and posterior tibial artery)”.
May you better explain the application and technique for readers? Did you inject ACT along the occluded below-the-knee vessel(s) or only along the “wound related artery”? Did you inject ACT also in the main foot vessels such as pedal artery and/or medial and plantar arteries according to the wound angiosome area? Did you perform local perilesional administration in all patients included?
Methods have been extensively revised in order to comply with the reviewer’s (well-taken) criticisms (lines 153-157).
- If Authors have also tested ABI at the baseline, may you report the main values at the enrolment?
ABI was tested as per local routinely protocol of the Clinic; however, we had chosen not to report this data due to a high number of patients with severe medium-calcinosis of the arteries.
- It has been reported that TcPO2 is recorded at the baseline on the 1st toe and at the ankle (do you mean in rearfoot in the area of perfusion of posterior tibial artery?).
We confirm the TcPO2 has been recorded in the area of perfusion of posterior tibial artery (see line 136-137).
In addition, TcPO2 was evaluated during the follow-up only on the 1st toe.
May you better explain your procedure? Why did you perform TcPO2 also at the ankle if t80% of patients had forefoot DFUs? or you mean only for patients with heel ulcers?
This is a consolidated procedure of this Clinic and the rationale is to explore both TA and TP area of perfusion. The value used for the primary endpoint is that of 1st toe (as reported in Methods ‘Primary endpoints’ and Results). We now have been modified Table 1 adding this data.
During the follow why did you perform TcPO2 only on the first toe also for patients with heel uclers?
The majority of patients have a forefoot ulcer and therefore we decided to use the 1st toe values for the primary endpoint (the analyses excluded patients with a hindgfoot ulcer; this has been now steted in the manuscript). For the only two patients with an hind-foot ulcer, TcPO2 values at the ankle level have been analyzed, with a trend toward increase of TcPO2 values at any time-point (see lines 233-235, Table 1 and 3).
Results
In the text you reported that TcPO2 values are reported in table 2 and cost analysis in table 3. Please revise, TcPO2 values are now reported in table 3 and costs in table 2.
Done! Well taken! Thanks
In table 1, could you report the ulcers size at the baseline?
Done
Could you report what kind of surgical interventions the patients underwent after ACT? i.e toe amputation, transmetatarsal amputation, Lisfranc or Chopart amputation, sequestrectomy, calcanectomy, etc. It can help to understand the severity of DFUs and the burden of surgical procedures.
Information has been added in new Tab. 1.
- Could you report (again in table 1) how many patients had a desert foot with the absence both of pedal and plantar arteries?
Done
Discussion
Some limitation should be added to the text such as the absence of a control group which probably is the main limitation. In addition, there is not post-procedural angiogram evaluation (not ethical of course) to evaluate the real and the grade of improvement of SAD after ACT. Nonetheless, this element does not influence the validity of the study.
Done (see last par.)
Round 2
Reviewer 2 Report
The authors have answered all the questions adequately. They have adequately justified their reasons for setting out the study as they have and I agree with them. Further research and confirmation of the method's success with a well-designed prospective randomised study is definitely needed.
As far as the discussion is concerned, the suggestions have been duly taken on board and the additional references suggested have also been added. They have also fulfilled the requirement and provided conclusions in an appropriate format.
I consider that the revised article is suitable for publication.